# Practical high-dimensional quantum key distribution protocol over deployed multicore fiber

Mujtaba Zahidy[1], Domenico Ribezzo[2,3,4], Claudia De Lazzari[5], Ilaria Vagniluca[5], Nicola Biagi[5], Ronny Müller[1], Tommaso Occhipinti[5], Leif K. Oxenløwe [1], Michael Galili[1], Tetsuya Hayashi [6], Dajana Cassioli [7,8], Antonio Mecozzi [2,8], Cristian Antonelli[2,8], Alessandro Zavatta[3,5] & Davide Bacco [5,9] ✉

Quantum key distribution (QKD) is a secure communication scheme for sharing symmetric cryptographic keys based on the laws of quantum physics, and is considered a key player in the realm of cyber-security. A critical challenge for QKD systems comes from the fact that the ever-increasing rates at which digital data are transmitted require more and more performing sources of quantum keys, primarily in terms of secret key generation rate. High-dimensional QKD based on path encoding has been proposed as a candidate approach to address this challenge. However, while proof-of-principle demonstrations based on lab experiments have been reported in the literature, demonstrations in realistic environments are still missing. Here we report the generation of secret keys in a 4-dimensional hybrid time-path-encoded QKD system over a 52-km deployed multicore fiber link forming by looping back two cores of a 26-km 4-core optical fiber. Our results indicate that robust high-dimensional QKD can be implemented in a realistic environment by combining standard telecom equipment with emerging multicore fiber technology.

Global data transmission is increasing exponentially every year[1] and security of communication against new forms of attacks has become a relevant concern for scientific communities and cyber-security experts, as well as for the general public. While computationally secure encryption protocols[2,3] are the basic building blocks of today's communication network, quantum key distribution (QKD) is the only known method that can guarantee information-theoretic secure encryption between two or more parties. In particular, QKD allows to share a symmetric cryptographic key, useful to encrypt data communication without binding the security to any assumption on the attacker's capabilities. The two most important figures of merit of QKD

are the secret-key generation rate, and the overall link distance between users, which are both limited by the loss and noise levels affecting the transmitted quantum signals. Indeed, in the absence of high-performance quantum memory[4], which is still challenging today[5], the achievable rate and distance for a given error rate affecting the quantum signal reception are limited, with bounds discussed in[6].

An alternative approach to addressing the noise limit and the low generation rate relies on the use of high-dimensional (Hi-D) quantum states—*qudits*—for quantum communication purposes. As shown in[7–9], protocols based on qudits are considerably more resilient to noise than protocols based on qubits. These protocols allow the generation

[1]Department of Electrical and Photonics Engineering, Technical University of Denmark, Ørsteds Pl., Kgs. Lyngby 2800, Denmark. [2]Department of Physical and Chemical Sciences, University of L'Aquila, L'Aquila, Italy. [3]Istituto Nazionale di Ottica, Consiglio Nazionale delle Ricerche (CNR-INO), Firenze 50125, Italy. [4]University of Naples Federico II, Napoli, Italy. [5]QTI S.r.l., Firenze 50125, Italy. [6]Optical Communications Laboratory, Sumitomo Electric Industries, Ltd., Yokohama 244-8588, Japan. [7]Department of Information Engineering, Computer Science and Mathematics, University of L'Aquila, L'Aquila, Italy. [8]National Laboratory of Advanced Optical Fibers for Photonics (FIBERS), CNIT, L'Aquila, Italy. [9]Department of Physics and Astronomy, University of Florence, Via Sansone 1, Firenze 50019, Italy. ✉e-mail: davide.bacco@unifi.it

of a secure key even at high noise levels and can provide high communication capacity. More in detail, the highest tolerance in two-dimensional systems based on the BB84 protocol using one-way reconciliation, in the case of coherent attacks, corresponds to a quantum bit error rate (QBER) of ~11%[10], whereas this threshold increases to 18% and 24% for four-dimensional and eight-dimensional systems, respectively[7,8].

Proof-of-concept experiments exploiting Hi-D encoding have already been demonstrated using different degrees of freedom such as frequency[11], orbital angular momentum (OAM)[12–14], path-encoding[15,16], OAM-Path hybrid[17], OAM-Polarization hybrid[18], time-energy and time-bin encoding[19–21] and Frequency-time hybrid[22]. Multicore fibers, whose potential in transmitting classical data at petabit-per-second rates has been recently demonstrated[23], are also being considered for quantum applications. The reason is that they allow implementing practical Hi-D QKD systems based on path-encoding protocols, which do not suffer from state fragility seen in other protocols like those based on OAM encoding, and they do not reduce the effective qudit rate−a drawback of protocols based on time-bin encoding. However, all demonstrations of Hi-D path-encoding reported in the literature are in-lab tests[15,24,25].

In this work, we report the successful transmission of hybrid time-path Hi-D quantum states with a final secret key rate (SKR) of 51.5 kbps through a 52-km long multicore fiber link exhibiting 22 dB of channel loss, deployed in the city of L'Aquila, Italy[26]. Our experiment paves the way towards the deployment of future Hi-D QKD systems in real scenarios.

## Results
### Experimental Implementation
We performed the 4D time-path-encoding QKD field trial experiment in a 52-km multicore fiber deployed in the city of L'Aquila, Italy. The fiber loop was characterized by 22 dB of losses, mostly due to fiber connectors. The implemented protocol is a 4-dimensional generalization of BB84, where weak coherent pulses form the quantum states. In order to detect photon-number splitting attacks, Alice and Bob run a two-decoy-state protocol, using states with unequal mean photon numbers $\mu_1$ and $\mu_2$, together with the vacuum state[27,28]. Alice transmits the signal states with probabilities $p_{\mu_1}$ and $p_{\mu_2}$, whose values result from the optimization of the secure key generation rate. The transmitted states are modulated in the $\mathcal{X}$ or $\mathcal{Z}$, the two mutual unbiased bases, with probabilities $P_{\mathcal{Z}}$ and $P_{\mathcal{X}}$, respectively. A detailed description of the protocol is provided in the Methods section.

Figure 1a is a plot of the expected secret key rates versus channel loss in the 4D protocols. The secret key rate is evaluated based on the experimental parameters of Table 1[29], where QBER$_{\mathcal{Z}_{\mu_n}}$ and QBER$_{\mathcal{X}_{\mu_n}}$ are the quantum bit-error rates measured for the $\mathcal{Z}$ and $\mathcal{X}$ bases, respectively, and for an average photon number $\mu_n$ ($n = 1, 2$), and $p_{gate}$ is the probability that a photon falls within the photodetection temporal gate. The values of $p_{\mu_1}$ and $p_{\mu_2}$, as well as those of $P_{\mathcal{Z}}$ and $P_{\mathcal{X}}$ resulted from the maximization of the secret key rate. Note that although the Hi-D protocols provide higher resilience against errors, the final achievable range is shorter, and the SKR drops to zero at lower channel losses. The reason for this behavior must be found in the fact that a higher-dimensional system uses more detectors (or more active time of a detector in hybrid time-bin protocols) than a protocol with a lower dimension, and this results in higher dark count rates and lower signal-to-noise ratio, see Supplementary Note 1. At the channel loss of 22 dB the secret key rate of the 4D protocol was estimated to be 51.5 kbps. The stability of the QKD implementation is illustrated in Fig. 1b, where we plot the QBER measured in the $\mathcal{Z}$ and $\mathcal{X}$ bases in a time window of 60 minutes. The figure shows greater stability in the $\mathcal{Z}$ basis, which indicates that the relative phase between pulses propagating in the different fiber cores is sufficiently stable to guarantee long-term operation. On the other hand, the relative phase between pulses propagating in the same fiber core is expected to be at least as stable. In fact, the large QBER values observed in the $\mathcal{X}$ basis are to be ascribed to mechanical vibrations present in the environment and affecting the fiber-based interferometer used for receiving $\mathcal{X}$-states. Note that these QBER spikes are absent in controlled laboratory environments[30] and can be improved by acting on the receiver setup (improved isolation). On the other hand, the Z-basis phase which corresponds to the full multicore fiber interferometer, was intact as it was only influenced by the natural phase drift in the fiber. We compared the 4D implementation with a 2D path-encoded implementation in the same setup. The results of this comparison are presented in Supplementary Note 1.

## Discussion
High-dimensional QKD protocols are introduced and designed for their higher capacity and resilience against noise. This, in turn, yields a higher secure-key generation rate compared to standard 2D counterparts for approximately similar QBER values[7,8]. However, depending on the physical properties that are used to encode information, the practical implementation of a Hi-D QKD system may result in fragile quantum states and poor performance (e.g. in terms of QBER). For instance, qudits based on the orbital angular momentum (OAM) of

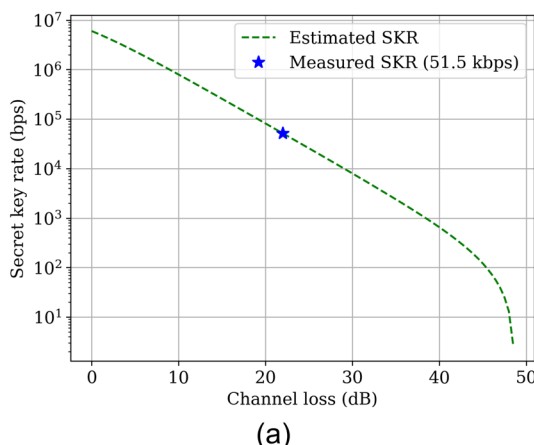

(a)

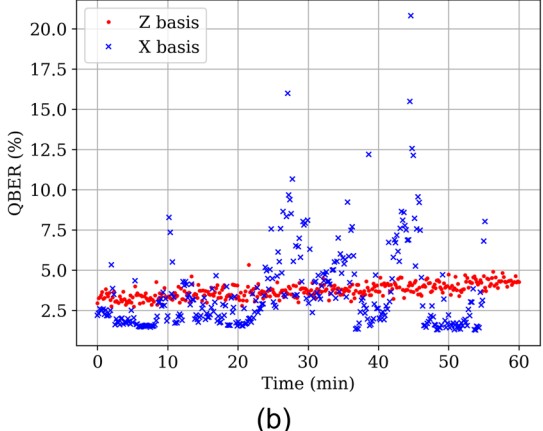

(b)

**Fig. 1 | Experimental results. a** Secret key generation rate as a function of the channel loss for the parameters of the implementation. The star represents the achieved generation rate at the field-deployed channel loss. **b** Long-term stability of the implementation. Each point represents 10 seconds of measurement. While the $\mathcal{Z}$ basis shows great stability over time - an indication of slow phase drift in the fiber −the $\mathcal{X}$ basis shows moments of high QBER due to vibration at the receiver site.

light suffer from crosstalk in optical fiber and turbulence in free space. In this work, through a field trial in a realistic urban environment, we demonstrate that path-encoding based on the use of multicore fibers is a robust scheme, standing out as a promising candidate for future Hi-D QKD systems. The robustness of this scheme is a consequence of the slow phase drift in deployed multicore fibers, which can be effectively tracked with the PLL developed for the field trial.

It is important to stress that, while in this work 4D states were formed in a time-path Hilbert space, in principle it is possible to generate Hi-D states with path-encoding alone. The advantage of avoiding time-bin encoding is an increase in quantum-state-generation rate, however, this approach requires stabilizing the phase of a larger number of spatial paths. Photonic integrated circuits (PICs) are an effective platform for the implementation of path-encoded Hi-D state sources and receivers[31,32], and are intrinsically scalable to large dimensions. For comparison, we listed multiple Hi-D studies indicating relevant parameters and the final SKR in Supplementary Note 2. Additionally, we added to the comparison some of the recent works in Supplementary Note 3 with normalized SKR per source repetition rate and the number of detectors with their efficiencies. This aims to provide a fair comparison taking into account only the protocol, its implementation, and the receiver's efficiency.

To summarize, we have demonstrated a Hi-D QKD system based on path-and-time-bin-encoding in a 52-km-long multicore fiber link field-deployed in the historical downtown area of the city of L'Aquila, Italy. Our results show higher noise tolerance of the proposed Hi-D scheme, compared to a reference 2D scheme, see Supplementary Note 1, as well as an enhanced secret key generation rate. This work paves the way toward the practical implementation of future Hi-D QKD systems.

## Methods

A complete description of the experimental setup is presented in supplementary note 4.

### Source

A detailed representation of the source structure is given in Fig. 2 -Alice box. A continuous wave laser, labeled as $Q$, is carved to form a train of 120 ps-long time-bin pulses with two cascaded intensity modulators, shown as one in the diagram, for a high extinction ratio. The state's repetition rate is 487 MHz with the two pulses of a time-bin state being separated in time by 800 ps. The carving follows a pseudo-random binary sequence of length $l = 2^{12}-1$, which is used to emulate random quantum state generation. Carving the pulses out of a CW laser increases the chance of phase correlation between consecutive qudits. It is necessary to devise a phase randomization stage to close the loophole and prevent attacks[33–35]. A second continuous wave laser, labeled as $M$, is attenuated and multiplexed with the quantum signal through a beam-splitter, and it serves as a monitor laser for the dual-band phase locking scheme. The two outputs of the beam-splitter impinge upon two electro-optical modulators (EOMs) that encode information into four-dimensional quantum states. To optimize the effect of the EOMs on the quantum signal and prevent undesired modulation in the monitor laser, the polarization of the quantum

**Table 1 | Experimental parameters**

| Parameter | 4D |
|---|---|
| $\mu_1$ ($\mu_2$) | 0.36 (0.16) |
| $p_{\mu_1}$ ($p_{\mu_2}$) | 0.81 (0.11) |
| $QBER_{Z_{\mu_1}}$ | 2.15% |
| $QBER_{Z_{\mu_2}}$ | 2.43% |
| $QBER_{X_{\mu_1}}$ | 3.54% |
| $QBER_{X_{\mu_2}}$ | 3.88% |
| $p_{gate}$ | 0.53 |

Measured QBERs, and achieved key rates in both bases, for the transmitted bases in the 4D protocol. The measured channel loss is 22 dB.

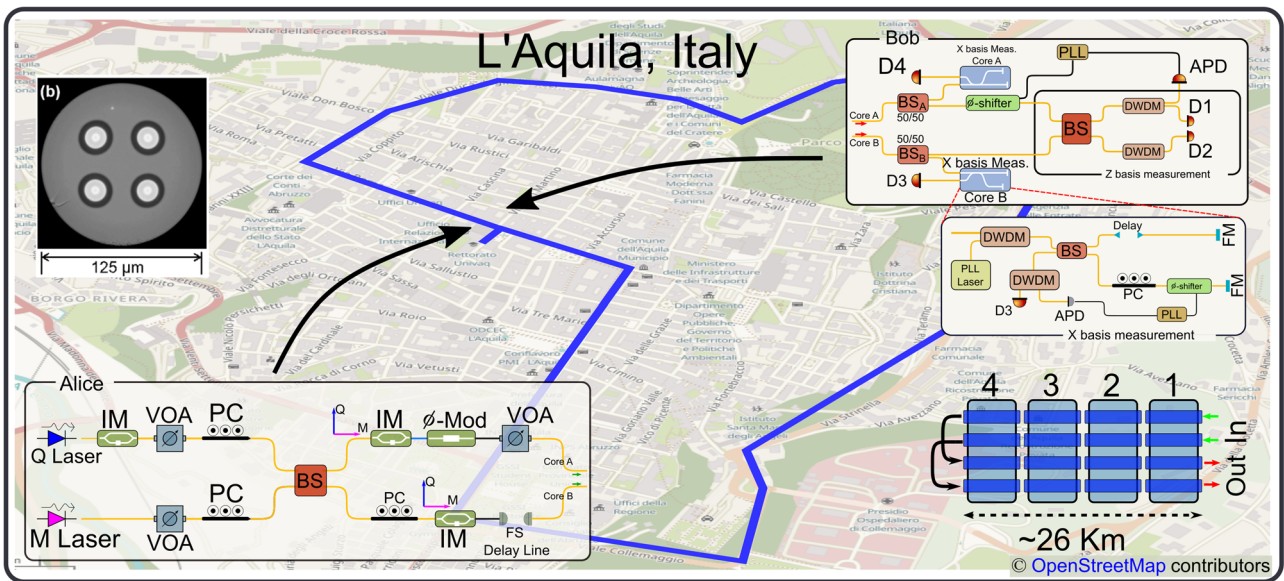

**Fig. 2 | Illustration of the setup.** Schematic of transmitter (Alice), receiver (Bob) both located at the University of L'Aquila headquarters, and multicore fiber channel. *Transmitter.* Q (M) Laser Quantum (Monitor) Laser, IM Intensity modulator, VOA Variable Optical Attenuator, PC Polarization Controller, $\phi$-Mod Phase Modulator, BS Beam-splitter, FS Free-Space. *Receiver.* PLL Phase-Locked Loop, $\phi$-shifter Piezo Phase Shifter, DWDM Dense Wavelength Division Multiplexer, APD Avalanche Photo-Diode, FM Faraday Mirror. *Channel.* Channel is formed by concatenating four multicore fibers with four uncoupled cores, each of approximately 6.5-km length. At the endpoint, two cores are connected back into the others, resulting in a 52-km-long two-core fiber link. The multicore fibers are deployed in an underground tunnel in the historical downtown area of the city of L'Aquila, Italy. The map depicting the city of L'Aquila and the span of the multicore fiber is roughly drawn based on the exact map presented in ref. 26. **b** The cross-section of the MCF. The four cores and the trench design are visible in the photo[26].

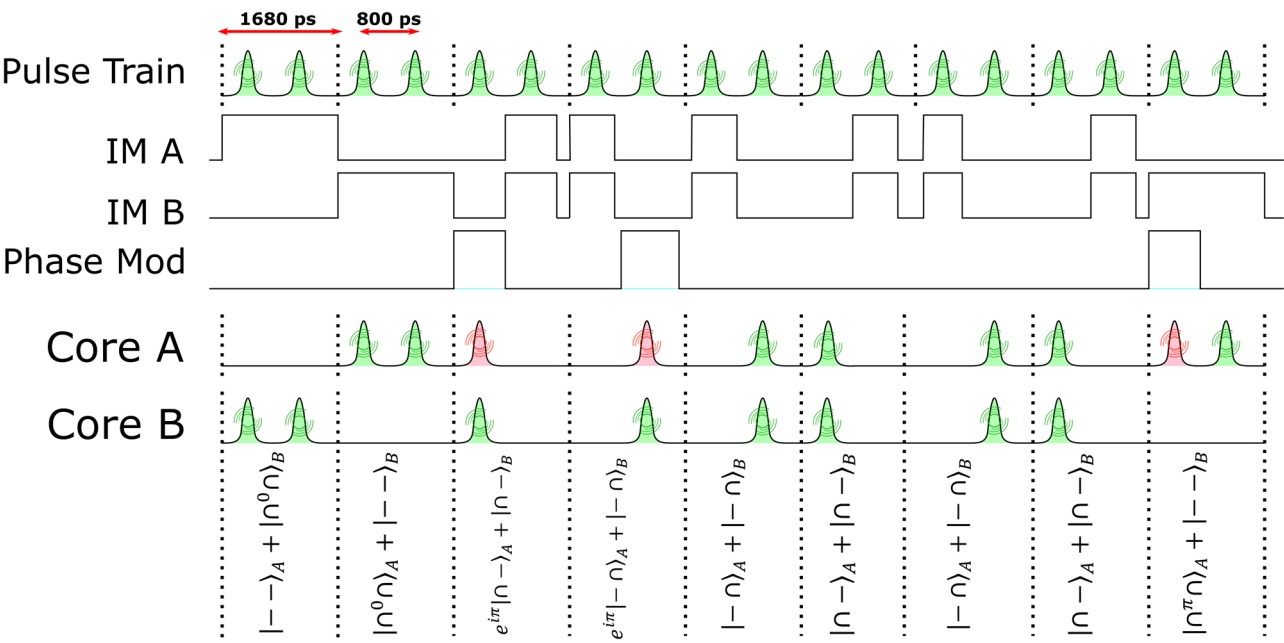

**Fig. 3 | High dimensional state generation.** Two replicas of a periodic train of pulse pairs are intensity-modulated at the output of a beam-splitter, prior to transmission in fiber cores A and B, while only the replica in the path to core A is also phase-modulated, so as to generate states in the $\mathcal{Z}$ and $\mathcal{X}$ bases. An RF signal is used to synchronize modulation in the two spatial paths, where the modulation signals are produced by a field-programmable gate array. Examples of the signals transmitted in two fiber cores are shown in the lower part of the figure, along with the corresponding quantum states.

signal is aligned with the EOM ordinary axis and that of the monitor laser with the extraordinary axis. The polarization alignment is achieved by means of the three polarization controllers shown in the figure. Insertion loss in the two spatial paths is equalized with the use of a variable optical attenuator, while a free-space delay line is used to equalize the corresponding propagation delays and maximize visibility in the reception of $\mathcal{Z}$-states.

The state encoding process is illustrated in Fig. 3. The train of pulse pairs, with each pulse forming a time-bin state, is intensity-modulated at the output of the beam-splitter so that either a $\mathcal{Z}$-state or an $\mathcal{X}$-state is generated in each time slot. For the former, one pulse of a pair of subsequent pulses is suppressed in each spatial path, whereas for the latter, both pulses of a pair of subsequent pulses are suppressed in one of the two spatial paths. A phase modulator then applies a phase $\phi \in \{0, \pi\}$ to generate $|\cap^{\phi}\cap\rangle_A + |--\rangle_B$.

All the EOM modulators are driven by a field-programmable gate array that also provides a down-sampled clock signal for synchronization to both the transmitter and receiver.

### Phase stabilization
We employed the dual-band phase stabilization technique to compensate for relative phase drifts between the two fiber cores. The monitor signal, launched at a frequency 400 GHz lower than that of the quantum signal, is co-propagated with the quantum signal and experiences similar phase changes. At the receiver (for any basis choice), two cascaded dense wavelength division multiplexers (DWDMs), each with 25–30 dB isolation, are used to demultiplex the quantum and monitor/PLL signals, and to effectively suppress out-of-band noise impairing the quantum signal. The avalanche photodiodes produce suitable signals for the PLL circuits with a monitor-signal power of approximately - 45 dBm, which is the value targeted with the monitor signal launch power used in the experiment. The PLL features an ADuC7020 micro-controller unit (MCU) from Analog Devices with four 12-bit analog to digital converters (ADCs) and 41.78 MHz clock rate. It produces a feedback signal that is used to control a phase shifter, as can be seen in Fig. 2. Details on the PLL design can be found in ref. 36.

### Receiver
The receiver (Bob) is composed of two $2 \times 2$ beam-splitters, $BS_A$ and $BS_B$, placed in each core to passively select the basis of measurement. One output of each beam-splitter is used for $\mathcal{X}$-basis measurements, while the other two outputs are used for $\mathcal{Z}$-basis measurements. The $\mathcal{X}$-basis receiver scheme is implemented in the form of a fiber-based unbalanced Michelson interferometer, with 800 ps temporal delay in one arm.

One output of each beam-splitter enters an unbalanced Michelson interferometer that implements the $\mathcal{X}$-basis measurements, while the other two outputs are used for $\mathcal{Z}$-basis measurements.

The unbalanced Michelson interferometer is fiber-based, with 800 ps temporal delay in one arm. A PLL with an additional laser source at the same frequency of the monitor laser is used to compensate for unavoidable phase drifts. The monitor laser signal transmitted with the quantum signal is too weak to be used in the PLL and does not interfere with the locally added laser signal. (PLL laser in Fig. 2). Two DWDMs are used to multiplex and demultiplex the PLL laser and the $\mathcal{X}$-signal state to be received. Other parts of the interferometer are two Faraday mirrors used to compensate for polarization rotations due to fiber propagation, and a free-space delay element, needed to compensate for differential propagation delays in the two arms. the overall loss of the interferometer is around 4 dB.

The $\mathcal{Z}$-basis measurement is performed by interfering signals in the two cores on a beam-splitter, and the quantum signal is extracted by means of two DWDMs prior to detection. The combined DWDMs for isolation add 1.5 dB of loss. The monitor signal from one of the two DWDMs feeds an APD to provide feedback to the PLL circuit.

All the detectors used for the quantum signals are superconducting nanowire single-photon detectors.

### Protocol
Here, we describe the protocol that allows Alice and Bob to establish a common secure key using 4D time-path encoding.
- Alice prepares the states described in the "section Source" and sends them through the quantum channel. The bases are chosen randomly with probabilities $P_{\mathcal{Z}}$ and $P_{\mathcal{X}}$, and the states are

prepared also randomly with no memory. As a countermeasure against photon-number splitting attacks, a decoy-state protocol is implemented, where the mean-photon number of each quantum state is adjusted to two pre-selected values with probabilities $p_{\mu_1}$ and $p_{\mu_2}$.

- Bob randomly selects a basis and performs the measurement. In our implementation, a beam-splitter passively chooses the basis of measurement.
- After a round of transmission and measurement, Alice and Bob communicate the basis of state preparation and measurement. At this stage, they discard all the instances in which the preparation and measurement bases mismatch.
- Out of the remaining instances of exchanged qudits, Alice and Bob publicly reveal part of the prepared states and measurement outcomes to estimate the quantum bit error rate.
- The remaining qudits yield two bits of information per detection instance, on which error reconciliation and privacy amplification are applied, ending with the establishment of a shared secure key.

### Quantum channel

The protocol implemented in this work expands the Hilbert space dimension of a $2D$ time-bin protocol into a $4D$ space by adding the path dimension. The protocol requires two optical paths or channels to encode information in the relative phase of the two paths. In our implementation, the spatial paths are the cores of four 6.5-km long uncoupled-core four-core optical fibers deployed in the city of L'Aquila, Italy. The characteristics of the individual cores are similar to those of standard single-mode fibers, with negligible crosstalk between cores (below $-40$ dB/km)[26,37]. Each fiber is terminated by a spliced SC MCF connector on both ends. This allows to concatenate all the available fibers and form a 26-km fiber link. The resulting quantum channel length is 52 km, since only two cores are used for the path encoding, allowing to connect and loop back two cores into the other two. The overall fiber loss is measured to be 22 dB, of which 12 dB are to be ascribed to the SC MCF connectors and the fan-in/fan-out connectors.

### Parameter optimization

In order to maximize the gain in this implementation, we optimized the 4D final secret key rate, Eq. (1) where the parameters are given in Table 1.

The secret key length $\ell_{4D}$ is estimated in a finite-block-size regime with secrecy parameter $\epsilon_{\text{sec}}$, correctness parameter $\epsilon_{corr}$, and post-processing block size $n$ in 4D with[38]

$$\ell_{4D} \leq 2D_0^Z + D_1^Z[2 - H(\phi_Z)] - \lambda_{EC} - 6\log_2{(21/\epsilon_{\text{sec}})} - \log_2(2/\epsilon_{corr}), \quad (1)$$

where $D_0^Z$ and $D_1^Z$ are the lower bounds of vacuum and single-photon events in the $Z$ basis, $H(x) := -x\log_2(x/3) - (1-x)\log_2(1-x)$ is the Shannon entropy for 4D variables, $\phi_Z$ is the phase error rate upper bound, and $\lambda_{EC}$ is the number of bits that are publicly disclosed during error correction[38]. The term $-6\log_2{(21/\epsilon_{\text{sec}})}$ bounds the information that Eve has about the key, while $-\log_2{(2/\epsilon_{corr})}$ corresponds to the bit disclosed during error verification; $\epsilon_{sec} = \epsilon_{corr} = 10^{-12}$ have been set. We chose a block size of $n_Z = 10^8$ for finite key analysis for 4D protocol, and we estimated for the 4D states an error reconciliation efficiency $f_{err,4D} = \lambda_{EC}/(n_Z * H_{4D}(\phi_Z)) = 1.06$, which is in line with the most recent results reported in the literature for the measured QBER values adopting the original cascade error correction protocol[39].

Parameters such as mean photon numbers for signal and decoy state, $\mu_1$ and $\mu_2$ respectively, probabilities of transmitting signal and decoy, $p_{\mu_1}$ and $p_{\mu_2}$, and probabilities of basis choice, $P_{\mathcal{X}}$ and $P_{\mathcal{Z}}$, are chosen to maximize the secret key rate. In the optimization, security

parameters, channel loss, detector efficiencies, and dark counts are also taken into account.

### Data availability

The data that support the findings of this study are available from the corresponding authors upon request.

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

## Acknowledgements
C.A. and A.M. are funded by QUID (Quantum Italy Deployment) funded by the European Commission in the Digital Europe Programme under the grant agreement No 101091408., L.O. received funding from the Center of Excellence SPOC (ref DNRF123), A.Z. received funding from the Project QuONTENT under the Progetti di Ricerca, CNR program funded by the Consiglio Nazionale delle Ricerche (CNR) and by the European Union—PON Ricerca e Innovazione 2014-2020 FESR—Project ARS01/00734 QUANCOM. D.B received funding from the IFD DK project Fire-Q (No. 9090-00031B), by EQUO (European Quantum Ecosystem) funded by the European Commission in the Digital Europe Programme under the grant agreement No 101091561 and by the European Union ERC project QOMUNE (101077917). The authors acknowledge OpenStreetMap[40] for providing the map of L'Aquila City, Italy.

## Author contributions
D.B., A.Z., and M.Z. conceived the research idea and designed the experiment. M.Z., D.R., A.Z., and D.B. performed the measurements; M.Z., D.R., C.D.L., I.V., N.B., R.M. performed the data analysis, M.Z., D.R., C.A., A.M., and D.B. wrote the manuscript. L.K.O., T.O., M.G., D.C. provided critical feedback together with all authors. T.H., C.A., A.M., and D.C. facilitate access to the fiber infrastructure.

## Competing interests
The authors declare no competing interests.
