## [Peer Review File · Nature Communications]

Practical High-Dimensional Quantum Key Distribution Protocol over deployed Multicore fiberEditorial Note: Parts of this Peer Review File have been redacted as indicated to remove third-party material where no permission to publish could be obtained.

REVIEWERS' COMMENTS:

Reviewer #1 (Remarks to the Author):

In the manuscript “Practical High-Dimensional Quantum Key Distribution Protocol over deployed Multicore fiber”, the authors propose a new path-encoding scheme for high-dimensional quantum key distribution and demonstrate an experimental implementation of 4-dimensional quantum key distribution over the field-installed multicore fiber link. The topic is very interesting and the experimental is extremely challenging. The authors have used the optimized encoding scheme and phase-locking techniques to overcome perturbations in the installed fiber and achieved, for the first time, high-dimensional QKD through the deployed multicore fiber link. This work represents a meaningful attempt and an important step forward in the development and application of high-dimensional QKD in the deployed quantum network.

The manuscript is well-written and easy to follow. The results are clearly presented with comprehensive description of the experimental details. I believe this manuscript is worthy of publication in Nature Communications.

Here are some specific comments which may help enhance the readability of this manuscript:

1. As described in the Introduction, the noise tolerance in the 4-dimensional QKD are higher than the 2-dimensional QKD. However, Fig. 3 a) shows that the simulated maximum working distance of the 4-dimensional QKD are shorter than the 2-dimensional QKD. It would be interesting to read an explanation about the case. Maybe it is related to the error rate during the quantum state preparing.
2. Fig. 3 b) shows the long-term stability of the system running continuously for an hour. It is easy to see that the Z basis error rate is relatively stable, while the X basis error rate

fluctuates significantly. Have similar fluctuations been observed for the secure key rate?

3. In Line 116-117, the authors described that Multicore fibers have unique features such as low relative phase and polarization drift and high core-to-core isolation. Are there any references?

4. Readers may be interested in certain experimental details, such as the overall efficiency of the receiver (without the detector efficiency) and the block size used for finite key processing. It would be helpful to specify these details in the Method section.

5. In the presented experiment, the 52-km multicore fiber link is realized by 26-km multicore fiber in a round-trip or loop-back manner. This feature should be specified in the Abstract.

6. The pulses are generated by carving a CW laser (Q laser) into a pulses train. It is generally believed that the phase between different pulses is correlated in this situation. Phase randomization is one of the key assumptions in the decoy-state quantum key distribution. May I ask if active phase randomization have been implemented in the experiment? If not, mention this inconspicuous limitation in the Discussion section would be better.

7. I noticed that the sending period of quantum states in this experiment is 1680ns, while the time interval between two time-bins in the quantum state is 800ps, which is not half of the period. Does this configuration increase any risk of time-shift attack?

8. In the Abstract, the authors claim the record-high secure key rate. However, direct comparison with previous work reported in the literature seems missing. Providing relevant information and direct comparison would help support this claim.

9. There maybe a few typos to be fixed and some changes to improve the presentation:

1) In Line 84, the author describes the link loss as 21dB. However, multiple places later in the manuscript indicate that the link loss is 22dB (such as in Line 207, in the caption of Table 1, in Line 226, and in the caption of Fig. 3). Perhaps there is a typo in Line 84 that needs to be corrected.

2) In the caption of Fig. 1, "... blocking 1 time-bin of time-bin states ..." should be "... blocking one time bin of time-bin states..." or "... blocking one bin of time-bin states ..."

3) In Line 315, the expression "2x2" use the letter "x" to represent the multiplication sign, which could be written more rigorously, as in Line 157.

4) It is recommended that the authors add a schematic diagram of the multicore fiber cross-section to Fig. 2, so that the reader can understand the characteristics of the quantum link

more clearly.

5) The title of the manuscript maybe shortened as “Practical High-Dimensional Quantum Key Distribution over deployed Multicore fiber”, which can summarize the main work more clearly, since this work covers both the protocol and the experimental implementation. It is only a suggestion.

Reviewer #2 (Remarks to the Author):

The authors report a record-high secret-key rate generation and distribution in a 4-dimensional path-encoded QKD system with more than 100% improvement, compared to standard QKD systems, in a 52-km deployed multicore fiber link. This result indicates that robust high-dimensional QKD can be implemented in a realistic environment by leveraging standard telecom equipment and multicore fiber technology.

However, through a detailed examination of the paper, we find that some of the authors' results are overstated. First, the authors' improvement is compared to their own system and not to the most advanced standard BB84 system. The authors simply run the BB84 and HD protocols separately in the same system and link, and then claim that the comparison of the two can achieve a 100% improvement in secret key rate. As far as I know, in a field-deployed optical fiber, the BB84 protocol can achieve a secret key rate of approximately 100 kbps@16.4 dB or 2.5 Mbps@3.9 dB [Npj Quantum Inf. 5, 101 (2019)], which is much better than the results obtained in the paper. Compared with some other high-dimensional systems, such as the work in Ref. [Sci. Adv. 3, e1701491 (2017)], the results obtained in the paper do not seem sufficient to be published in Nature Communications.

Other suggestions:

1. The author should disclose more experimental parameters in the results, including the number of pulses sent and some security parameters for key estimation.
2. It is hard to prove that the data results obtained from the experiment can match the simulation results well, as the author has only collected one data point. I suggest that the author can provide more data by increasing attenuation.
3. In order to highlight the contribution of this paper to the field, I suggest that the author

compare the results with previously published works, especially BB84 systems in field-link or other non-field HD systems, to show the significance of this work.

Reviewer #3 (Remarks to the Author):

Dear Editor,

I have read with great interest the manuscript entitled "Practical High-Dimensional Quantum Key Distribution over deployed Multicore Fiber". In this work, the authors address the research topic of long-distance transmission of high-dimensional quantum systems. In their implementation, the authors adopt the well-known hybrid encoding strategy, where different degrees of freedom of a single quantum system are used to define a larger Hilbert Space. The authors report on the propagation of a 4-dimensional quantum system propagated over 52km (26km) of a deployed fiber composed of several cores.

Unfortunately, I think the manuscript in its current form cannot be published. First, there are two important issues in this work that must be carefully analyzed by the authors: 1) there are several claims in the work that are not justified by the results presented (I discuss them next). And second, 2) the strategy used by the authors to highlight the relevance of their results in higher dimensions has an important flaw.

Last, I would also point out that the manuscript presentation and discussion must be greatly improved. Looking at Figure 2, for instance, it's impossible to properly understand the experimental setup used. I still don't understand where Alice is. From Fig 2 it seems it's far from Bob station, but I believe this is not the case. Also, the paper's subject has been extensively studied in the literature. Nonetheless, there are only 32 references and 40% (!!!!!!!) of them are auto-citations. A practice to be discouraged by editors in the future I hope.

The important points that in my opinion must be carefully considered are:

1) Invalid Claims

For instance, in the abstract and throughout the paper, the authors claim that they "...report a record high secret key generation rate and distribution in a 4-dimensional path-encoded QKD system with more than 100% improvement compared to standard QKD system...". To be very clear, this is a very unjustified claim.

First, the authors must clearly distinguish that they are working on a hybrid encoding strategy (There are plenty of works on this subject, please check them), and the propagated Hi-D system is not defined only in terms of the spatial degree of freedom. This is crucial here because later the authors used a different encoding to make the comparison with a bi-dimensional path-based QKD system. I further discuss this strategy next. But by no means is the path-based QKD system a standard scheme for QKD, and its generation rates are known to be low compared with what can be achieved with time-bin or polarization. So, when the authors claim that they get a 100% improvement of what is possible with standard QKD, this is very misleading.

2) Flaw Strategy to present the results

The authors implement two different schemes for doing QKD over the considered network. One is based on a bi-dimensional path-encoded quantum system. The second one is based on a hybrid 4-dimensional quantum system encoded in terms of the path and time-bin of a photon. The authors explain that both schemes were implemented almost simultaneously, such that a direct comparison between their QBER or Secret Key rate would "...provides a fair comparison between 2D and 4D QKD schemes."

Unfortunately, I strongly disagree with this point. While I understand the reasoning, I would say that the proper way to make such a comparison would be two possibilities: 1) Or the authors encode the high-dimensional system in the same degree of freedom, that is, using all the paths available in the fiber or, 2) The authors use the lower dimensional system that is known to be more robust and faster for the purpose of comparison, in this case, a bi-dimensional time-bin based QKD system.

Point 1 is easy to justify. I would expect that when more cores of the fibers are considered, the stabilization of the system will be more complex and the final 4-dimensional QKD system slower. Point 2 is also easy to justify. I would expect that the speed that can be achievable with a bi-dimensional time-bin based QKD system will be much higher than what was obtained with a path-encoded qubit. In this case, only one fiber core would be needed and there is no need to use any stabilization system. Results for schemes like this can be readily found in the literature.

So, in my opinion, the comparison performed has important flaws and must be corrected to properly highlight the advantages obtained in higher dimensions. Such a comparison may not even be necessary, but in the way the results are presented now, I cannot recommend the paper for publication.

I. REVIEWER 1

Comments to the Author: In the manuscript “Practical High-Dimensional Quantum Key Distribution Protocol over deployed Multicore fiber”, the authors propose a new path-encoding scheme for high-dimensional quantum key distribution and demonstrate an experimental implementation of 4-dimensional quantum key distribution over the field-installed multicore fiber link. The topic is very interesting and the experimental is extremely challenging. The authors have used the optimized encoding scheme and phase-locking techniques to overcome perturbations in the installed fiber and achieved, for the first time, high-dimensional QKD through the deployed multicore fiber link. This work represents a meaningful attempt and an important step forward in the development and application of high-dimensional QKD in the deployed quantum network.

The manuscript is well-written and easy to follow. The results are clearly presented with comprehensive description of the experimental details. I believe this manuscript is worthy of publication in Nature Communications.

Here are some specific comments which may help enhance the readability of this manuscript:

1. **As described in the Introduction, the noise tolerance in the 4-dimensional QKD are higher than the 2-dimensional QKD. However, Fig. 3 a) shows that the simulated maximum working distance of the 4-dimensional QKD are shorter than the 2-dimensional QKD. It would be interesting to read an explanation about the case. Maybe it is related to the error rate during the quantum state preparing.**

Reply: We would like to thank the reviewer for evaluating the manuscript with such attention. It is true that the noise tolerance of Hi-D QKD is higher. However, the complete description of noise-skr, presented in [1] and refs [40-42] therein, shows that SKR approaches zero for lower attenuation values of the channel as the protocol’s dimensionality increases. The reason for this behavior must be found in the fact that a higher-dimensional system uses more detectors (or more active time of a detector in time-bin protocols) than a lower-dimensional one, and this results in higher dark count rates. Essentially, the protocol become unsuccessful when the clicks due to dark counts increase the QBER beyond a certain threshold. Therefore, for high-dimensional

protocols, this occurs at lower attenuation values.

[1] High-Dimensional Quantum Communication: Benefits, Progress, and Future Challenges - Daniele Cozzolino, Beatrice Da Lio, Davide Bacco, Leif Katsuo Oxenløwe, Adv. Quantum Technol. **2019**, 2, 1900038

2. **Fig. 3 b) shows the long-term stability of the system running continuously for an hour. It is easy to see that the Z basis error rate is relatively stable, while the X basis error rate fluctuates significantly. Have similar fluctuations been observed for the secure key rate?**

Reply: The secret key rate is obtained by processing blocks of raw data with size 10^8 in which we included all the data acquired during the data acquisition. However, we excluded the blocks that were corresponding to the phase locking loop losing the locking condition as it was being monitored constantly. This was done by setting a threshold on the X-basis QBER. We would like also to mention that the loss of the locking point was due to mechanical vibration present in the environment which affected the fiber-based interferometer and can be improved by acting on the receiver setup (improved isolation). On the other hand, the Z-basis phase which corresponds to the full multi-core fiber interferometer, was intact as it was only influenced by the natural phase drift in the fiber.

3. **In Line 116-117, the authors described that Multicore fibers have unique features such as low relative phase and polarization drift and high core-to-core isolation. Are there any references?**

Reply: The phase stability between two cores of a multicore fiber compared to two single-mode fibers were measured and reported for the case of a 7-core 2-km long fiber [1], see Fig. 1. The cross-talk, in the fiber in use for this implementation, is reported in [2] to be more than 60 dB isolation in spool fiber and more than 55 dB for the deployed fiber, see Fig. 2. Relative polarization drift was measured and reported in [1] and [3] for more than 7000 seconds. We haven't performed a direct long-term tracking of relative polarization drift in multicore fiber. However, the phase stability results indicate high relative polarization stability during data acquisition. We expect any mode mismatch to affect the phase stability which was not observed during the experiment.

FIGURE REDACTED

FIG. 1. Stability comparison of a multicore fiber and two identical single mode fiber. Outputs of a balanced interferometer with 2 km long arms consisting of: two single-mode fibers bundled together (purple and pink traces), or two cores in the same multicore fiber (red and orange traces). The inset shows the phase drifts between the two single-mode fibers over one second. Counts are collected by two single-photon detectors integrated over 10 ms during an overall 30 s time window (y-axis with arbitrary units) [1]

FIGURE REDACTED

FIG. 2. The optical properties changes before and after cabling and installation [2].

[1] Stable Transmission of High-Dimensional Quantum States Over a 2-km Multicore Fiber - Beatrice Da Lio, et.al. - IEEE Journal of Selected Topics in Quantum Electronics, 26, 4 (2020)

[2] Field-Deployed Multi-Core Fiber Testbed - Tetsuya Hayashi; Takuji Nagashima; Tetsuya Nakanishi; Tetsu Morishima; Reiji Kawawada; Antonio Mecozzi; Cristian

Antonelli - 2019 24th OptoElectronics and Communications Conference (OECC) and 2019 International Conference on Photonics in Switching and Computing (PSC)

[3] Use of Optical Coherent Detection for Environmental Sensing - Antonio Mecozzi; Cristian Antonelli; Mikael Mazur; Nicolas Fontaine; Haoshuo Chen; Lauren Dal-lachiesa; and Roland Ryf - Journal of Lightwave Technology Vol. 41, Issue 11, pp. 3350-3357 (2023)

4. **Readers may be interested in certain experimental details, such as the overall efficiency of the receiver (without the detector efficiency) and the block size used for finite key processing. It would be helpful to specify these details in the Method section.**

Reply: We thank the reviewer for mentioning this point. The Z-basis measurement, which is a direct measurement of the time of arrival of the photons is only affected by the loss introduced due to DWDMs filtering out the monitor signal and the fiber connectors. We used two DWDMs to achieve maximum isolation. In total, it introduced approximately 1.5 dB of loss (0.5-0.6 per DWDM + 0.3-0.5 for connectors). The X-basis measurement, which is done with a fiber-based unbalanced Michelson interferometer, has an extra 4 dB of loss due to multiple components inside the interferometer. The chosen key block size is 10^8 , and has been added in the main text.

We added these details to the *Method* section.

5. **In the presented experiment, the 52-km multicore fiber link is realized by 26-km multicore fiber in a round-trip or loop-back manner. This feature should be specified in the Abstract.**

Reply: We thank the reviewer for this remark. We fixed the abstract to be more clear.

... compared to standard QKD systems, in a 52-km deployed multicore fiber link forming by looping back two cores of a 26-km 4-core fiber. Our results indicate that ...

6. **The pulses are generated by carving a CW laser (Q laser) into a pulses train. It is generally believed that the phase between different pulses is**

correlated in this situation. Phase randomization is one of the key assumptions in the decoy-state quantum key distribution. May I ask if active phase randomization have been implemented in the experiment? If not, mention this inconspicuous limitation in the Discussion section would be better.

Reply: We thank the reviewer for this point. Indeed we agree with the reviewer and our measurement also shows coherence between consecutive pulses that constitute consecutive qubits. A solution to this problem can be found in the patent [1], where a telecom laser is used in active modulation working above and below the threshold so the global phase of each quantum state will be random. We added a phrase in the discussion to stress the necessity of adding the phase randomization stage.

Carving the pulses out of a CW laser increases the chance of phase correlation between consecutive qudits. It is necessary to devise a phase randomization stage to close the loophole and prevent attacks [2, 3, 4].

[1] Quantum key distribution transmitter - Bacco, Davide and Cataliotti, Francesco Saverio and De Natale, Paolo and Occhipinti, Tommaso and Vagniluca, Ilaria and Zavatta, Alessandro - WO2022189523A1

[2] Security of quantum key distribution with imperfect devices - Gottesman, Daniel and Lo, H-K and Lutkenhaus, Norbert and Preskill, John - International Symposium on Information Theory, 2004. ISIT 2004. Proceedings, 136, 2004, IEEE

[3] Source attack of decoy-state quantum key distribution using phase information - Yan-Lin Tang, Hua-Lei Yin, Xiongfeng Ma, Chi-Hang Fred Fung, Yang Liu, Hai-Lin Yong, Teng-Yun Chen, Cheng-Zhi Peng, Zeng-Bing Chen, and Jian-Wei Pan - Phys. Rev. A **88**, 022308

[4] A Review of Security Evaluation of Practical Quantum Key Distribution System - Shihai Sun, Anqi Huang - Entropy 2022, 24(2), 260

7. I noticed that the sending period of quantum states in this experiment is 1680ns, while the time interval between two time-bins in the quantum state is 800ps, which is not half of the period. Does this configuration increase any risk of time-shift attack?

Reply: The states are generated by driving the carver with a signal generated by

the FPGA. The FPGA features a sampling rate of 12.5 GHz or 80 ps. We decided to use 21 sampling points for each qubit. The transition to an equidistributed pulse is straightforward, however, this configuration helps in characterizing the setup. To answer the question, we expect no information leakage as time is the degree of freedom in which the bit is encoded and cannot be revealed in a trusted measurement scenario. The time-shift attack expects a difference in the efficiency of detectors at different instances. In this scenario, each detector is allocated for one measurement basis. However, here we only use one detector for Z measurement (time of arrival measurement). On the other hand, the relative phase information is always measured on the late pulse, decoupling any correlation between the relative phase and timing distribution. In other words, there is no security problem related to the different time-interval.

8. **In the Abstract, the authors claim the record-high secure key rate. However, direct comparison with previous work reported in the literature seems missing. Providing relevant information and direct comparison would help support this claim.**

Reply: We would like to thank the reviewer for mentioning the details that improve the quality of the paper. Below, we presented a table of articles we found that were most relevant for comparison. Unfortunately, a fair and thorough comparison is always hard in the context of QKD as the parameters involved are too diverse. However, we tried to list the final SKR in the table as the final figure of merit. We would like to stress that in some of the works, the final SKR was low due to inefficient equipment, however, in some, the technique does not allow to achieve a high SKR. Furthermore, the categories of protocol span all the degrees of freedom and hybrid ones. To the best of our knowledge, one of the two fully path-encoding, ref [1] has a modulation rate of 1 KHz due to use of deformable mirrors and ref [12], a modulation of 5 KHz due to heater limitations. Most hybrid works in which path is one of the degrees of freedom use a path-OAM combination which dictates to be free-space, slow due to SLM, and short channels mainly on an optical table. Furthermore, in cases where the source is based on pair generation, the transmitter loss also affects the overall performance of the system.

Ref	ν [Hz]	D	Protocol	DoF	channel length [km]	η_{Ch} [dB]	η_{Rx} [dB]	η_{SPD} [dB]	QBER %	SKR [Kbps]
This work	487 M	4	Eff. BB84	Path-time	52	22	4	0.809	3.5	113
[1]	1 K	4	BB84	Path	0.3		24.5	12.2	10.25 ± 0.6	$(4.31 \pm 1.19) * 10^{-9}$
[2]*	30	16	BB84	OAM	0.0005				13.4 ± 4 15.6 ± 7	$7.5 * 10^{-6}$ $3.1 * 10^{-6}$
[3]	4 K	14	BB84	OAM	0.002		35.1	1.87	10.5	$6.8 * 10^{-3}$
[4]	297.6 M	4	BB84	TB	25 65 105 145	5.1 14 23 31.5	2.5	9.2	3.4 3.4 4.9 7.9	37 24 5.5 0.42
[5]		3	DO-QKD	Energy-time	242				7.9	$6 * 10^{-5}$
[6]	1 M	1024	photon-efficient HDQKD	Energy-time	20			0.458	39.6	2700
[7]	312.5 M 625 M 1.25 G	16 8 4	DO-QKD	Energy-time	– 41 43	0.1 7.6 12.7	6	1.67	6.5 4.8 4.9	23000 5300 1200
[8] [†]	2.5 G	4	HD-QKD	TB	4 8 10 14 16.6	20 40 50 70 83		1.55	4.5	26200 11900 7710 3400 1070
[9]	60	4	HD-QKD	Path-OAM						
[10]				Energy-time					5	
[11]	80 M	4 5	HD-QKD	OAM	< 1			10	8.8 14	1.139 0.8606
[12]	5 K	4	BB84	Path		– 10 20 –	8		13	– – – –
[13]	600 M	4	HD-QKD	OAM	1.2	1.2	10		14-18	37.85

TABLE I. Comparison of recent studies on HD QKD in terms of source repetition rate, dimension, implemented protocol, channel loss, and the final SKR. All the η values are presented as loss factors in the system and should be regarded as negative.

[†] The SKR drops rapidly to zero after 18 dB.

* The final SKR is not reported. This number corresponds to sifted key rate.

- [1] Cañas, G., et al. "High-dimensional decoy-state quantum key distribution over multicore telecommunication fibers." *Physical Review A* 96.2 (2017): 022317
- [2] Etcheverry, Sebastian, et al. "Quantum key distribution session with 16-dimensional photonic states." *Scientific reports* 3.1 (2013): 2316.
- [3] Mirhosseini, Mohammad, et al. "High-dimensional quantum cryptography with twisted light." *New Journal of Physics* 17.3 (2015): 033033.
- [4] Vagniluca, Ilaria, et al. "Efficient time-bin encoding for practical high-dimensional quantum key distribution." *Physical Review Applied* 14.1 (2020): 014051.
- [5] Liu, Jingyuan, et al. "High-dimensional quantum key distribution using energy-time entanglement over 242 km partially deployed fiber." *arXiv preprint arXiv:2212.02695* (2022).
- [6] Zhong, Tian, et al. "Photon-efficient quantum key distribution using time-energy entanglement with high-dimensional encoding." *New Journal of Physics* 17.2 (2015): 022002.
- [7] Lee, Catherine, et al. "Large-alphabet encoding for higher-rate quantum key distribution." *Optics express* 27.13 (2019): 17539-17549.
- [8] Islam, Nurul T., et al. "Provably secure and high-rate quantum key distribution with time-bin qudits." *Science advances* 3.11 (2017): e1701491.
- [9] Efficient High-Dimensional Quantum Key Distribution with Hybrid Encoding - Yonggi Jo, Hee Su Park, Seung-Woo Lee, and Wonmin Son - *Entropy* 2019, 21(1), 80
- [10] Large-Alphabet Quantum Key Distribution Using Energy-Time Entangled Bipartite States - Irfan Ali-Khan, Curtis J. Broadbent, and John C. Howell - *Phys. Rev. Lett.* **98**, 060503
- [11] Higher-dimensional orbital-angular-momentum-based quantum key distribution with mutually unbiased bases - Mhlambululi Mafu, Angela Dudley, Sandeep Goyal, Daniel Giovannini, Melanie McLaren, Miles J. Padgett, Thomas Konrad, Francesco Petruccione, Norbert Lütkenhaus, and Andrew Forbes - *Phys. Rev. A* **88**, 032305
- [12] High-dimensional quantum key distribution based on multicore fiber using silicon photonic integrated circuits - Yunhong Ding, Davide Bacco, Kjeld Dalgaard, Xinlun

Cai, Xiaoqi Zhou, Karsten Rottwitt & Leif Katsuo Oxenløwe - *npj Quantum Information* volume 3, Article number: 25 (2017)

[13] Fiber-based high-dimensional quantum communications - Davide Bacco, Beatrice Da Lio, Daniele Cozzolino, Yunhong Ding, Michael Galili, Karsten Rottwitt, Leif K. Oxenløwe

9. There maybe a few typos to be fixed and some changes to improve the presentation:

- (a) In Line 84, the author describes the link loss as 21dB. However, multiple places later in the manuscript indicate that the link loss is 22dB (such as in Line 207, in the caption of Table 1, in Line 226, and in the caption of Fig. 3). Perhaps there is a typo in Line 84 that needs to be corrected.
- (b) In the caption of Fig. 1, "... blocking 1 time-bin of time-bin states ..." should be "... blocking one time bin of time-bin states..." or "... blocking one bin of time-bin states ..."
- (c) In Line 315, the expression "2x2" use the letter "x" to represent the multiplication sign, which could be written more rigorously, as in Line 157.
- (d) It is recommended that the authors add a schematic diagram of the multicore fiber cross-section to Fig. 2, so that the reader can understand the characteristics of the quantum link more clearly.
- (e) The title of the manuscript maybe shortened as "Practical High-Dimensional Quantum Key Distribution over deployed Multicore fiber", which can summarize the main work more clearly, since this work covers both the protocol and the experimental implementation. It is only a suggestion.

Reply: We thank the reviewer for mentioning the errors. We fixed them in the revised edition and added more details in the figure.

II. REVIEWER 2

Comments to the Author: The authors report a record-high secret-key rate generation and distribution in a 4-dimensional path-encoded QKD system with more than 100% improvement, compared to standard QKD systems, in a 52-km deployed multicore fiber link. This result indicates that robust high-dimensional QKD can be implemented in a realistic environment by leveraging standard telecom equipment and multicore fiber technology.

However, through a detailed examination of the paper, we find that some of the authors' results are overstated. First, the authors' improvement is compared to their own system and not to the most advanced standard BB84 system. The authors simply run the BB84 and HD protocols separately in the same system and link, and then claim that the comparison of the two can achieve a 100% improvement in secret key rate. As far as I know, in a field-deployed optical fiber, the BB84 protocol can achieve a secret key rate of approximately 100 kbps@16.4 dB or 2.5 Mbps@3.9 dB [Npj Quantum Inf. 5, 101 (2019)], which is much better than the results obtained in the paper. Compared with some other high-dimensional systems, such as the work in Ref. [Sci. Adv. 3, e1701491 (2017)], the results obtained in the paper do not seem sufficient to be published in Nature Communications.

Reply: We would like to thank the reviewer for evaluating our manuscript. Regarding the point on the comparison of QKD systems, we believe that a fair comparison would take into account all the parameters affecting the final key rate. Of those, the source rate, detection efficiency (Detector's technology, Single photon detection efficiency, dead-time, etc.), receiver's loss, and configuration e.g. the number of single photon detectors used for the specific experiment, channel loss, and if multiplexing is employed, should be considered to have a meaningful and fair comparison of the final SKR, at least in the context of QKD based on weak coherent pulses. While we believe it is very crucial to find ways to increase the rate, whether by using higher rate source or efficient detectors such as [1, 2, 3, 4], it does not lessen any merit from researches and efforts that try to explore newer aspects as a solution to other problems. With this regard, the first work mentioned by the reviewer is at GHz regime with the second paper with an effective qubit rate of 625 MHz and 70% detectors' efficiency. As mentioned in the paper, the SKR drops to zero at 19 dB of loss. Comparing the two implementations, 2D and 4D, on the same field test guarantees similar values affecting the result. We chose the 2D path-encoding over the 2D time-bin in our test

FIG. 3. Comparison of the achieved rates and the rate obtained in the lab with the efficient three-states BB84, time-bin encoding.

since we expected better results due to lower X-basis QBER. However, we added the data point for 2D time-bin in the similar channel loss and same detectors, taken in the lab, which you can find below.

[1] GHz detection rates and dynamic photon-number resolution with superconducting nanowire arrays - Giovanni V. Resta, et.al., arXiv:2303.17401

[2] Fast single-photon detectors and real-time key distillation enable high secret-key-rate quantum key distribution systems - Fadri Grünenfelder, et.al. - Nature Photonics volume **17**, pages 422–426 (2023)

[3] A 16-Pixel Interleaved Superconducting Nanowire Single-Photon Detector Array With A Maximum Count Rate Exceeding 1.5 GHz - Weijun Zhang, et.al. - IEEE Transactions on Applied Superconductivity, Volume: 29, Issue: 5, August 2019

[4] Performance and security of 5 GHz repetition rate polarization-based quantum key distribution - Fadri Grünenfelder, et.al. - Appl. Phys. Lett. **117**, 144003 (2020)

In table II we compare the mentioned works with our results, by normalizing the secure key rate based on the number of detectors used in the key-extraction base and the state generation rate at the source. In fig. 4 we plot the same data to provide an idea of how the

normalized secure key rate changes with increasing channel attenuation.

Ref.	ν (Hz)	Protocol	channel (dB)	N_{Detz}	SKR (bps)	SKR/ ν/N_{Det}
[1]	2.5 G	efficient BB84	2.5	16	115.8 M	0.002895
			19.6		2.6 M	6.5e-5
			55.1		233	5.825e-9
[2]	2.5 G	3 states BB84	1.58	14	64 Mbps	0.00183
			16.34		3Mbps	8.57e-5
[3]	2.5 G	HD QKD (4D)	4	4	26.2	0.00262
			10		7.71	0.000771
			16.6		1.07	0.000107
This work	487 M	BB84	22	2	23.6	2.422e-5
		HD QKD			51.5	5.286e-5

TABLE II. Caption

[1] Li, Wei, et al. "High-rate quantum key distribution exceeding 110 Mb s⁻¹." Nature Photonics 17.5

[2] Grünenfelder, Fadri, et al. "Fast single-photon detectors and real-time key distillation enable high secret-key-rate quantum key distribution systems." Nature Photonics 17.5 (2023): 422-426.

[3] Islam, Nurul T., et al. "Provably secure and high-rate quantum key distribution with time-bin qudits." Science advances 3.11 (2017): e1701491.

1. **The author should disclose more experimental parameters in the results, including the number of pulses sent and some security parameters for key estimation.**

Reply: The qubit rate is reported to be 595 MHz starting from a 1.19 GHz pulse rate with each two pulses forming a time-bin qubit distributed in 2 paths. However, due to signal preparation mismatches, some of the states were not formed according to the protocol resulting in an effective rate of 487 MHz. We performed the analysis with the correct source rate, however, in the first version of the manuscript, the source rate was mistakenly reported before correction due to signal mismatches. The loss

FIG. 4. Comparison with different works. The secure key rates in the y axis are normalized per the source generation rate and the number of detectors in the key generation basis. The x axis express the attenuation of the utilized channel.

in the transmitter is compensated by setting the power of the laser and attenuation properly such that the total mean photon number per qubit is according to Table 1 in the manuscript. The channel is formed by concatenating the four ≈ 6.5 km long 4-core fibers. Each fiber demonstrates 0.201–0.246 dB/km of loss at 1550 nm [1]. Each fiber has a fan in/out with an SC connector. The total end-to-end loss of the 52 km fiber is measured to be 22 dB. The receiver, as depicted in the figure *setup*, is formed by three interferometric measurements. In the Z-basis, after the beamsplitter, two DWDMs filter the quantum signals and phase monitoring laser, contributing to the loss of 1-1.5 dB per beamsplitter arm. In the X-basis, the interferometer shows close to 4 dB of loss. Finally, we reported extra information about the postprocessing steps.

We added the missing details to the manuscript:

$\epsilon_{sec} = \epsilon_{corr} = 10^{-12}$ have been set. We chose a block size of $n_Z = 10^8$ for both the 2D and 4D protocol, and we estimated for the 4D states an error reconciliation efficiency $f_{err,4D} = \lambda_{EC}/(n_Z * H_{4D}(\phi_Z)) = 1.06$, which is in line with the most recent results reported in the literature for the measured QBER values adopting the original cascade error correction protocol [2]. In the 2D protocol, $f_{err,2D} = \lambda_{EC}/(n_Z * H_{2D}(\phi_Z)) = 1.08$ has been adopted [3].

[1] Field-Deployed Multi-Core Fiber Testbed - Tetsuya Hayashi; Takuji Nagashima; Tetsuya Nakanishi; Tetsu Morishima; Reiji Kawawada; Antonio Mecozzi; Cristian Antonelli - 2019 24th OptoElectronics and Communications Conference (OECC) and 2019 International Conference on Photonics in Switching and Computing (PSC)

[2] Efficient Information Reconciliation for High-Dimensional Quantum Key Distribution - Ronny Mueller and Domenico Ribezzo and Mujtaba Zahidy and Leif Katsuo Oxenløwe and Davide Bacco and Søren Forchhammer - arXiv:2307.02225, 2023

[3] High performance reconciliation for practical quantum key distribution systems - Hao-Kun Mao, Qiong Li, Peng-Lei Hao, Bassem Abd-El-Atty & Abdullah M. Ilyasu - Optical and Quantum Electronics volume 54, 163 (2022)

It is hard to prove that the data results obtained from the experiment can match the simulation results well, as the author has only collected one data point. I suggest that the author can provide more data by increasing attenuation.

Reply: We agree that more data points would make matching the experimental data and simulation easier. Unfortunately, 52 km was the maximum distance available and we didn't want to add artificial loss in the channel as it does not encapsulate the main challenge of Hi-D encoding based on path-encoding, and phase fluctuation.

In order to highlight the contribution of this paper to the field, I suggest that the author compare the results with previously published works, especially BB84 systems in field-link or other non-field HD systems, to show the significance of this work.

FIG. 5. Comparison of the achieved rates and the rate obtained in the lab with the efficient three-states BB84, time-bin encoding.

Reply: We thank the referee for the question, which gives us the opportunity to enhance our work. We neglect the works that our either proof of concept, or too old to implement fast-rate technologies. For example [1 2 3] show a bit rate ranging from a few bits per hour up to the order of bit per second over very short channels, due to the slow quantum state generation rate. We show the most relevant results in a table.

Ref	ν [Hz]	D	Protocol	DoF	channel length [km]	η_{Ch} [dB]	η_{Rx} [dB]	η_{SPD} [dB]	QBER %	SKR [Kbps]
This work	483 M	4	Eff. BB84	Path-time	52	22	4	0.809	3.5	113
[1]	1 K	4	BB84	Path	0.3		24.5	12.2	10.25 ± 0.6	$(4.31 \pm 1.19) * 10^{-9}$
[2]*	30	16	BB84	OAM	0.0005				13.4 ± 4 15.6 ± 7	$7.5 * 10^{-6}$ $3.1 * 10^{-6}$
[3]	4 K	14	BB84	OAM	0.002		35.1	1.87	10.5	$6.8 * 10^{-3}$
[4]	297.6 M	4	BB84	TB	25 65 105 145	5.1 14 23 31.5	2.5	9.2	3.4 3.4 4.9 7.9	37 24 5.5 0.42
[5]		3	DO-QKD	Energy-time	242				7.9	$6 * 10^{-5}$
[6]	1 M	1024	photon-efficient HDQKD	Energy-time	20			0.458	39.6	2700
[7]	312.5 M 625 M 1.25 G	16 8 4	DO-QKD	Energy-time	– 41 43	0.1 7.6 12.7	6	1.67	6.5 4.8 4.9	23000 5300 1200
[8] [†]	2.5 G	4	HD-QKD	TB	4 8 10 14 16.6	20 40 50 70 83		1.55	4.5	26200 11900 7710 3400 1070
[9]	60	4	HD-QKD	Path-OAM						
[10]				Energy-time					5	
[11]	80 M	4 5	HD-QKD	OAM	< 1			10	8.8 14	1.139 0.8606
[12]	5 K	4	BB84	Path		– 10 20 –	8		13	– – – –
[13]	600 M	4	HD-QKD	OAM	1.2	1.2	10		14-18	37.85

TABLE III. Comparison of recent studies on HD QKD in terms of source repetition rate, dimension, implemented protocol, channel loss, and the final SKR. All the η values are presented as loss factors in the system and should be regarded as negative.

[†] The SKR drops rapidly to zero after 18 dB.

* The final SKR is not reported. This number corresponds to sifted key rate.

- [1] Cañas, G., et al. "High-dimensional decoy-state quantum key distribution over multicore telecommunication fibers." *Physical Review A* 96.2 (2017): 022317
- [2] Etcheverry, Sebastian, et al. "Quantum key distribution session with 16-dimensional photonic states." *Scientific reports* 3.1 (2013): 2316.
- [3] Mirhosseini, Mohammad, et al. "High-dimensional quantum cryptography with twisted light." *New Journal of Physics* 17.3 (2015): 033033.
- [4] Vagniluca, Ilaria, et al. "Efficient time-bin encoding for practical high-dimensional quantum key distribution." *Physical Review Applied* 14.1 (2020): 014051.
- [5] Liu, Jingyuan, et al. "High-dimensional quantum key distribution using energy-time entanglement over 242 km partially deployed fiber." *arXiv preprint arXiv:2212.02695* (2022).
- [6] Zhong, Tian, et al. "Photon-efficient quantum key distribution using time-energy entanglement with high-dimensional encoding." *New Journal of Physics* 17.2 (2015): 022002.
- [7] Lee, Catherine, et al. "Large-alphabet encoding for higher-rate quantum key distribution." *Optics express* 27.13 (2019): 17539-17549.
- [8] Islam, Nurul T., et al. "Provably secure and high-rate quantum key distribution with time-bin qudits." *Science advances* 3.11 (2017): e1701491.
- [9] Efficient High-Dimensional Quantum Key Distribution with Hybrid Encoding - Yonggi Jo, Hee Su Park, Seung-Woo Lee, and Wonmin Son - *Entropy* 2019, 21(1), 80
- [10] Large-Alphabet Quantum Key Distribution Using Energy-Time Entangled Bipartite States - Irfan Ali-Khan, Curtis J. Broadbent, and John C. Howell - *Phys. Rev. Lett.* **98**, 060503
- [11] Higher-dimensional orbital-angular-momentum-based quantum key distribution with mutually unbiased bases - Mhlambululi Mafu, Angela Dudley, Sandeep Goyal, Daniel Giovannini, Melanie McLaren, Miles J. Padgett, Thomas Konrad, Francesco Petruccione, Norbert Lütkenhaus, and Andrew Forbes - *Phys. Rev. A* **88**, 032305
- [12] High-dimensional quantum key distribution based on multicore fiber using silicon photonic integrated circuits - Yunhong Ding, Davide Bacco, Kjeld Dalgaard, Xinlun

Cai, Xiaoqi Zhou, Karsten Rottwitt & Leif Katsuo Oxenløwe - *Quantum Information* volume 3, Article number: 25 (2017)

[13] Fiber-based high-dimensional quantum communications - Davide Bacco, Beatrice Da Lio, Daniele Cozzolino, Yunhong Ding, Michael Galili, Karsten Rottwitt, Leif K. Oxenløwe

III. REVIEWER 3

Comments to the Author: I have read with great interest the manuscript entitled “Practical High-Dimensional Quantum Key Distribution over deployed Multicore Fiber”. In this work, the authors address the research topic of long-distance transmission of high-dimensional quantum systems. In their implementation, the authors adopt the well-known hybrid encoding strategy, where different degrees of freedom of a single quantum system are used to define a larger Hilbert Space. The authors report on the propagation of a 4-dimensional quantum system propagated over 52km (26km) of a deployed fiber composed of several cores.

Unfortunately, I think the manuscript in its current form cannot be published. First, there are two important issues in this work that must be carefully analyzed by the authors: 1) there are several claims in the work that are not justified by the results presented (I discuss them next). And second, 2) the strategy used by the authors to highlight the relevance of their results in higher dimensions has an important flaw.

Last, I would also point out that the manuscript presentation and discussion must be greatly improved. Looking at Figure 2, for instance, it’s impossible to properly understand the experimental setup used. I still don’t understand where Alice is. From Fig 2 it seems it’s far from Bob station, but I believe this is not the case. Also, the paper’s subject has been extensively studied in the literature. Nonetheless, there are only 32 references and 40% (!!!!!!!) of them are auto-citations. A practice to be discouraged by editors in the future I hope.

Reply to comment: We would like to thank the reviewer for evaluating our manuscript and for the comments that helped us improve its quality. We provided answers to points (1) and (2) below. Here, we add some clarifications. Regarding the schematic, in multiple instances, we pointed out that the fiber is looped back and indicated, on the bottom-right sketch of the fiber, the input and output are on the same side of the fiber. We changed the figure and added arrows to help the reader to have a better understanding of the system and implementation.

The experiment is performed in a loop-back implementation where Alice is connected to F_{in}^1 and Bob to F_{out}^4 where F^i is the i -th fiber. As depicted in the bottom-right corner of

Figure 2, we concatenated the fibers as below,

$$F_{out}^1 \rightarrow F_{in}^2 \quad F_{out}^2 \rightarrow F_{in}^3 \quad F_{out}^3 \rightarrow F_{in}^4 \quad (1)$$

to form a fiber from $F_{in}^1 \rightarrow F_{out}^4$. Finally, we connected two cores of F_{out}^4 to the other 2 cores to loop back. All the ≈ 6.5 km 4-core fibers are in the same bundle and started and terminated in the same server room. They are deployed in an underground tunnel, depicted in blue on the map, with approximately the same length.

Regarding citations, we added more relevant work to project a better variety of efforts in this field. However, to the best of our knowledge, the cited paper is the only and most recent work in the particular context. For example:

- Chip-based Multiplexing in multi-core fiber
- Study of phase stabilization in the deployed multi-core fiber
- High dimensional path-encoding
- Chip-based High dimensional QKD
- High Dimensional OAM in Air-core fiber
- High-dimensional Entangled OAM in Air-core fiber

We added the following phrases, in the caption of Fig. 2 to the manuscript and hope this clears out the confusion.

Schematic of transmitter (Alice), receiver (Bob), both located at the University of L'Aquila headquarters ...

The important points that in my opinion must be carefully considered are:

1. Invalid Claims

For instance, in the abstract and throughout the paper, the authors claim that they "... report a record high secret key generation rate and distribution in a 4-dimensional path-encoded QKD system with more than 100% improvement compared to standard QKD system...". To be very clear, this is a very unjustified claim.

FIGURE REDACTED

FIG. 6. Schematic of transmitter (Alice), receiver (Bob) both located at the University of L'Aquila headquarters, and multicore fiber channel. **Alice.** Q (M) Laser: Quantum (Monitor) Laser, IM: Intensity modulator, VOA: Variable Optical Attenuator, PC: Polarization Controller, ϕ -Mod: Phase Modulator, BS: Beam-splitter, FS: Free-Space. **Bob.** PLL: Phase-Locked Loop, ϕ -shifter: Piezo Phase Shifter, DWDM: Dense Wavelength Division Multiplexer, APD: Avalanche Photo-Diode, FM: Faraday Mirror. **Channel.** Channel is formed by concatenating four multicore fibers with four uncoupled cores, each of approximately 6.5-km length. At the endpoint, two cores are connected back into the others, resulting in a 52-km-long two-core fiber link. The multicore fibers are deployed in an underground tunnel in the historical downtown area of the city of L'Aquila, Italy. b) The cross-section of the MCF. The four cores and the trench design are visible in the photo [ref].

First, the authors must clearly distinguish that they are working on a hybrid encoding strategy (There are plenty of works on this subject, please check them), and the propagated Hi-D system is not defined only in terms of the spatial degree of freedom. This is crucial here because later the authors used a different encoding to make the comparison with a bi-dimensional path-based QKD system. I further discuss this strategy next. But by no means is the path-based QKD system a standard scheme for QKD, and its generation rates are known to be low compared with what can be achieved with time-bin or polarization. So, when the authors claim

that they get a 100% improvement of what is possible with standard QKD, this is very misleading.

Reply:

We thank the reviewer for his/her comment. We added the word *hybrid* in the abstract and introduction to distinguish it from the fully path-encoding implementation throughout the manuscript, and the protocol is clearly discussed.

We agree that path encoding is not a standard scheme for QKD as it requires a new type of fiber. However, it has its advantages over time-based encoding. Using a different degree of freedom other than time will preserve the source rate while in time-based encoding, the main motivation, generating a higher secret key rate, fades away due to occupying more bins for one qudit. The second motivation, resilience against noise, is not the dominant factor in low to moderate loss as the SNR allows to have below the allowed threshold QBER in 2D implementations. Also, we respectfully disagree with the statement “*But by no means is the path-based QKD system a standard scheme for QKD, and its generation rates are known to be low compared with what can be achieved with time-bin or polarization*”. The Hi-D encoding, regardless of the degree of freedom, generates higher SKR compared to 2D implementation when the source rate and QBER are equal and SNR is high enough. The main contribution to SNR is the detectors’ dark counts in high-loss channels. The effort in Hi-D QKD is focused on generating high-fidelity states and faithful transmission of them. Path-encoding allows the latter with phase stabilization techniques common also in Twin-Field QKD. Our claim “*Here we report a record-high secret-key rate generation and distribution in a 4-dimensional path-encoded QKD system with more than 100% improvement, compared to standard QKD systems, in a 52-km deployed multicore fiber link.*” clearly states **52 km of fiber or approximately 20 dB of channel loss**. First of all, we would like to stress that the loss is not due to fiber as it is measured to be 0.201-0.246 dB/km [1] and it mostly comes from the SC connectors. Secondly, 20 dB of loss is considered moderate channel loss where SNR is nowhere near the dark counts of the detectors. Hence, if QBER can be kept low, the higher generation is totally feasible. The QBER of the 2D implementation is comparable to other implementations at our channel loss. We would like to complete the argument by pointing out the claim

FIG. 7. Comparison of the achieved rates and the rate obtained in the lab with the efficient three-states BB84, time-bin encoding.

“Above 100% generation”. The generation rate is a result of the ratio of transmitting the key generation base (Z) and security check base (X). We ran a simulation to find the best ratio for our channel length and achieved QBER and set the parameters of the experiment accordingly.

The following changes are made in the manuscript to make it more clear;

In the abstract: Here we report a record-high secret-key rate generation and distribution in a 4-dimensional hybrid time-path-encoded QKD system

In the introduction: In this work, we report the successful transmission of hybrid time-path Hi-D quantum states through a 52-km long multicore fiber

Finally, we reported in this letter (fig. 7) a comparison of the achieved results with the results we got in the laboratory with the same BB84 setup, but encoding the states in the time-bin degree-of-freedom. The secure key rate for a channel loss of 22 dB with the time-bin encoding is absolutely comparable.

[1] Field-Deployed Multi-Core Fiber Testbed - Tetsuya Hayashi; Takuji Nagashima; Tetsuya Nakanishi; Tetsu Morishima; Reiji Kawawada; Antonio Mecozzi; Cristian Antonelli - 2019 24th OptoElectronics and Communications Conference (OECC) and 2019 International Conference on Photonics in Switching and Computing (PSC)

2. Flaw Strategy to present the results

The authors implement two different schemes for doing QKD over the considered network. One is based on a bi-dimensional path-encoded quantum system. The second one is based on a hybrid 4-dimensional quantum system encoded in terms of the path and time-bin of a photon. The authors explain that both schemes were implemented almost simultaneously, such that a direct comparison between their QBER or Secret Key rate would “..provides a fair comparison between 2D and 4D QKD schemes.”.

Unfortunately, I strongly disagree with this point. While I understand the reasoning, I would say that the proper way to make such a comparison would be two possibilities: 1) Or the authors encode the high-dimensional system in the same degree of freedom, that is, using all the paths available in the fiber or, 2) The authors use the lower dimensional system that is known to be more robust and faster for the purpose of comparison, in this case, a bi-dimensional time-bin based QKD system.

Point 1 is easy to justify. I would expect that when more cores of the fibers are considered, the stabilization of the system will be more complex and the final 4-dimensional QKD system slower. Point 2 is also easy to justify. I would expect that the speed that can be achievable with a bi-dimensional time-bin based QKD system will be much higher than what was obtained with a path-encoded qubit. In this case, only one fiber core would be needed and there is no need to use any stabilization system. Results for schemes like this can be readily found in the literature.

So, in my opinion, the comparison performed has important flaws and must be corrected to properly highlight the advantages obtained in higher dimensions. Such a comparison may not even be necessary, but in the way the results are presented now, I cannot recommend the paper for

FIG. 8. State Generation

publication.

Reply: We appreciate the point carefully raised by the reviewer. However, we believe that Figure 8 may address part of his/her concern. The figure shows that every pair of pulses forms a **Qudit**. The same is true for the 2D time-bin implementation where 2 consecutive pulses, whether they are sent or not, form a **qubit**. With this explanation, the *quantum state rate* of the two sources is similar and equal to half the source pulse rate, and any technique that improves one equivalently improves the other one as well. However, multiplexing in 2 cores (running 2 parallel 2D time-bin QKD) will improve the overall rate. Similarly, the qubit rate can be increased in 2D path encoding since each bin of the 4D implementation can be regarded as one qubit where information is encoded in either the path or the relative phase of the two paths. We, however, didn't make that comparison as it would compare two QKD transmitters with different rates. To complete the discussion, here we report the SKR of the 2D time-bin encoding in 22 dB channel loss and similar repetition rate and detectors.

We agree that time-bin is more robust although no difference in qubit generation rate is expected (without SDM). However, it also needs stabilization of the interferometer at the receiver. Furthermore, the only additional loss factor in 2D path-encoding compared to time-bin is due to WDMs to filter out the monitor signal from quantum one. During our experiment, the time-bin interferometer was showing instabilities due to uncontrolled environmental vibrations, while the path interference showed much stronger stability, as highlighted in Figure 9-Z basis measurement. Considering this point, if the 2D time-bin were implemented, we expected higher QBER resulting in

FIG. 9. **Stability of QBERs indicating phase stability in the implementation.**

lower SKR. In the graph, we added the data points of a 2D time-bin implementation.

Regarding the stabilization point, we already performed the 4-core stabilization and it is reported in [1]. The stabilization is completely classical without imposing any overhead to the quantum communication such as using a header or part of the qubits for drift/clock compensations [2] and hence does not affect the rate of the 4D system other than extra loss due to filters.

[1] Characterization and stability measurement of deployed multicore fibers for quantum applications - Davide Bacco, Nicola Biagi, Ilaria Vagniluca, Tetsuya Hayashi, Antonio Mecozzi, Cristian Antonelli, Leif K. Oxenløwe, and Alessandro Zavatta - Photonics Research Vol. 9, Issue 10, pp. 1992-1997 (2021)

[2] Simple quantum key distribution with qubit-based synchronization and a self-compensating polarization encoder - Costantino Agnesi, et.al. - Optica Vol. 7, Issue 4, pp. 284-290 (2020)

REVIEWERS' COMMENTS

Reviewer #1 (Remarks to the Author):

I have reviewed the response letter and the revised manuscript. I am pleased to see that my previous concerns have been adequately addressed, and the overall quality of the manuscript has improved. I would like to recommend the acceptance of this manuscript for publication in Nature Communications.

I would like to raise a minor issue regarding the Supplementary Materials. While Supplementary Materials are an essential component of the manuscript, I have observed that the current version contains references that are not consistently and rigorously cited. To meet the high standards of a journal like Nature Communications, I suggest that the authors take a more meticulous approach to revise and enhance the references within the Supplementary Materials section.

Reviewer #2 (Remarks to the Author):

The author's response is not very convincing to me, especially in regard to the author's statement about the record-breaking key rate and the 100% improvement of the secure key rate, which is the aspect I'm most concerned about. The author uses a rather mystical normalization method for comparison, including normalizing the repetition frequency of experiments and the number of detectors. I find it hard to accept this approach. After all, comparing experimental results should not be based on a theoretical comparison. I also wonder why the author didn't consider the number of optical fibers used, as compared to the BB84 protocol, the HD protocol requires more optical fibers. Therefore, I still believe this work is not sufficient to be published in Nature Communications and is more suitable for a specialized journal.

Reviewer #3 (Remarks to the Author):

Dear Editor,

I have seen the reply provided by the authors. They have addressed several of the points I have raised in my initial report. Even though I think there are still some minor points open on the current version of the paper, I am happy with the new version prepared by the authors. So, I am glad to recommend the paper for publication.